# Inference-Optimized High-Performance Photoelectric Target Detection Based on GPU Framework

**Shicheng Zhang [1], Laixian Zhang [2,*], Huichao Guo [2], Yonghui Zheng [3], Song Ma [4] and Ying Chen [4]**

[1] Graduate School, Space Engineering University, Beijing 101416, China
[2] Department of Electronic and Optical Engineering, Space Engineering University, Beijing 101416, China
[3] Beijing Space Information Transmission Center, Beijing 102300, China
[4] Southwest China Institute of Electronic Technology, Chengdu 610036, China
[*] Correspondence: zhanglaixian@pku.edu.cn; Tel.: +86-1366-133-8085

**Abstract:** Deep learning has better detection efficiency than typical methods in photoelectric target detection. However, classical CNNs on GPU frameworks consume too much computing power and memory resources. We propose a multi-stream inference-optimized TensorRT (MSIOT) method to solve this problem effectively. MSIOT uses knowledge distillation to effectively reduce the number of model parameters by layer guidance between CNNs and lightweight networks. Moreover, we use the TensorRT and multi-stream mode to reduce the number of model computations. MSIOT again increases inference speed by 9.3% based on the 4.3–7.2× acceleration of TensorRT. The experimental results show that the model's mean average accuracy, precision, recall, and F1 score after distillation can reach up to 94.20%, 93.16%, 95.4%, and 94.27%, respectively. It is of great significance for designing a real-time photoelectric target detection system.

**Keywords:** photoelectric targets; knowledge distillation; TensorRT acceleration; multi-stream; inference optimized

## 1. Introduction

The rapid development of optoelectronic equipment such as pinhole and miniature cameras has brought serious risks of information leakage, leading to serious property damage and threatening personal safety. In response to this adverse phenomenon, this work exploits the cat-eye effect to detect the camera and effectively solve it. For clarity, it should be mentioned that the cat-eye effect [1] is the incident-light irradiation of the focal plane photoelectric sensor inside the camera, forming a photoelectric target. Regarding the cat-eye effect, photoelectric target detection using active laser imaging [2] is currently the most effective way to detect the camera. This method focuses on the camera's physical characteristics.

Researchers in this field initially focused on detecting the optical properties of optoelectronic targets [3–7]. However, the detection effect was not ideal. After 2010, AlexNet [8], VGGNet [9], and ResNet [10] have made breakthroughs in the field of image recognition tasks, such as target detection, classification, face recognition, pose estimation, and other aspects. Krizhevsky et al. [8] designed a seven-layer AlexNet network to prove the effectiveness of convolutional neural networks (CNN), which has multi-level feature extraction ability [11,12], demonstrating an object detection effect that outperformed classical detection algorithms. As a result, researchers gradually focused on CNN. Some researchers used deep learning to extract photoelectric target feature information [13,14]. Ke [15] designed a fully automatic camera detection and recognition system based on the PC, which combines machine learning and neural network methods to identify surveillance camera equipment effectively. This method improved VGGNet-16, and the single forward inference time reached 5.36 s, which could not meet the real-time detection requirements of photoelectric targets and was unsuitable for engineering applications. Liu et al. [16]

developed a convolutional neural network photoelectric target detection and recognition system based on NVIDIA Jeston TX2, which uses a lightweight network to detect miniature indoor cameras. This method needs to improve the accuracy and inference acceleration of lightweight networks and relies too much on the computing performance of NVIDIA Jeston TX2. Moreover, Huang et al. [17] designed an improved YOLOv3 model based on the PC, which recognizes miniature cameras in a single frame. This method requires large-volume hardware support, and eliminating false targets in complex background environments needs to be more thorough, as this results in unstable detection accuracy.

Classical CNNs running on GPU computing frameworks [18–20] create problems with poor real-time performance. Therefore, this paper proposes a multi-stream inference-optimized TensorRT (MSIOT) inference acceleration method. This method verifies the real-time detection of photoelectric targets on the GPU computing frame with limited computing power. This paper's significant contributions are as follows:

- In order to effectively reduce the number of model parameters and ensure the high accuracy of the neural network, we trained a high-precision CNN model through knowledge distillation [21]; guided learning was performed on lightweight networks. Finally, a high-precision lightweight network model was obtained.
- In order to reduce the number of computations in the process of model inference, we deeply explored the inference acceleration principle of the TensorRT [22–24] engine based on the characteristics of the GPU computing framework and built a computational graph based on the existing network. Experiments verified the effectiveness and practicality of the TensorRT inference acceleration.
- In order to solve the excessive waste of time for data replication and overlapping calculations during the model inference, we optimized TensorRT to exploit CUDA (Computer Unified Device Architecture) control based on the kernel execution principle. The utilization of the GPU was fully invoked through the multi-stream [25] mode, further shortening the inference time of deep learning models.

## 2. Materials

### 2.1. Dataset

The datasets in this experiment included exclusively accurate shots. During the experiment, highly reflective substances such as glass, tin foil, plastic bottles, and metals were added to simulate false targets and increase the background complexity. For this study, we acquired 3000 active and 3000 passive images adopting the strategy described in the literature [14]. To obtain richer miniature camera target feature information, both images are 2048 × 2048 pixels high-resolution grayscale images. The dataset was collected and labeled in the active image (Figure 1) based on the characteristics of the active image enhancement target information. Specifically, the miniature camera lens' reflected flare (green area in the image) is the dataset's true target. The dataset's false targets are the metal, plastic film, and other items near the cabinet reflected flare (red areas in the image). We cropped 3000 active images acquired in various indoor scenarios and created 6570 datasets of true targets. Moreover, to enhance the neural network's object classification effect, the ratio of the number of real target and false target datasets is controlled to 1:1. Therefore, 6570 false target images are screened, each with a size of 20 × 20 pixels.

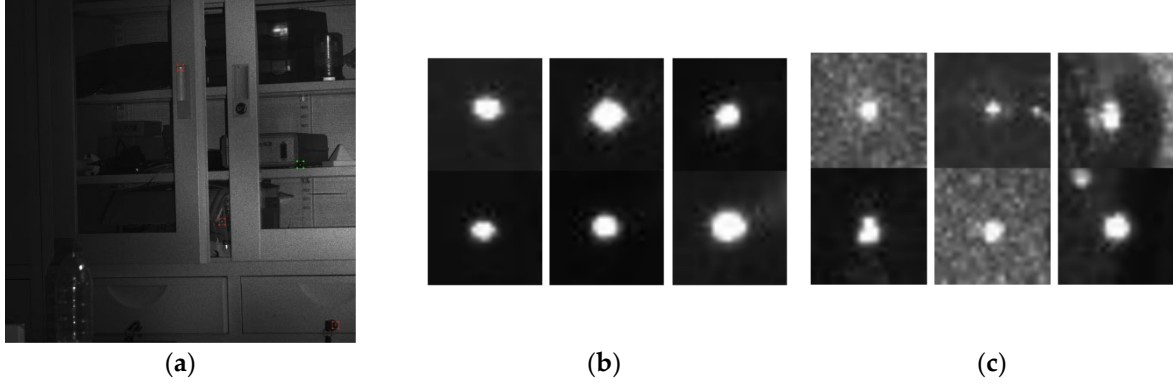

(**a**)　　　　　　　　　　　(**b**)　　　　　　　　　　　(**c**)

**Figure 1.** Sample images of a partial dataset: (**a**) active image labeled with a true target in the dataset, (**b**) part of the true target dataset, and (**c**) part of the false target dataset.

### 2.2. Experimental Environment

The proposed deep learning model and knowledge distillation process were implemented on Inter (R) Core (TM) i7-9700F CPU@3.00 GHz, GTX TITAN XP with 12 GB GPU memory, and a Windows 10 operating system with 64 GB system memory. The testing environment and TensorRT acceleration utilized a NVIDIA Jeston Nano 4 GB embedded development board, with a quad-core ARM Cortex-A57 MPCore processor and an NVIDIA Maxwell architecture GPU with 128 NVIDIA CUDA® cores, providing 472 GFlops of computing performance.

In the network training process, the optimizer was SGD, the cross-entropy loss function was BCELoss, the momentum was set to 0.9, the initial learning rate was set to 0.001, which dropped to the original 0.92 every ten generations, and the training samples for each learning were 30 for a total of 100 iterations. After completing the training, the best evaluation result was saved as the final model.

## 3. Methods

The main idea of this method is to ensure that the model's parameters and calculation burden are significantly reduced and its inference speed is improved while attaining high prediction accuracy. It is used to solve the problem of long running time and easy jamming caused by deploying a deep learning network model to a GPU computing framework with limited computing power. Therefore, we developed the MSIOT inference acceleration method, which fully considers the compatibility of model compression methods with inference acceleration. Knowledge distillation is adopted during model training, and multi-stream is used for optimization during model inference based on the TensorRT to build small memory engine files. It can further reduce the inference time to achieve fast real-time detection of photoelectric targets.

### 3.1. Knowledge Distillation

Many scholars have praised convolutional neural networks due to their excellent detection effects. However, directly deployed on the embedded device side, CNNs with a huge computation complexity will affect the operation, imposing poor real-time performance due to the device's hardware limitations. Given that lightweight networks have fast uptime but poor detection rates, this paper uses knowledge distillation between convolutional neural networks and lightweight networks to solve this problem effectively.

The complete knowledge distillation concept was first proposed in 2014 by Google Labs Hinton [24], which experimentally verified its feasibility and the effectiveness of CNN compression on MNIST datasets. The probability of an error class is relatively small in the probability distribution output of a well-trained photoelectric target detection model. Since its relative probability distribution hides the feature information that the real label does

not have, knowledge distillation is introduced to improve the discriminant accuracy of lightweight networks. As shown in Figure 2:

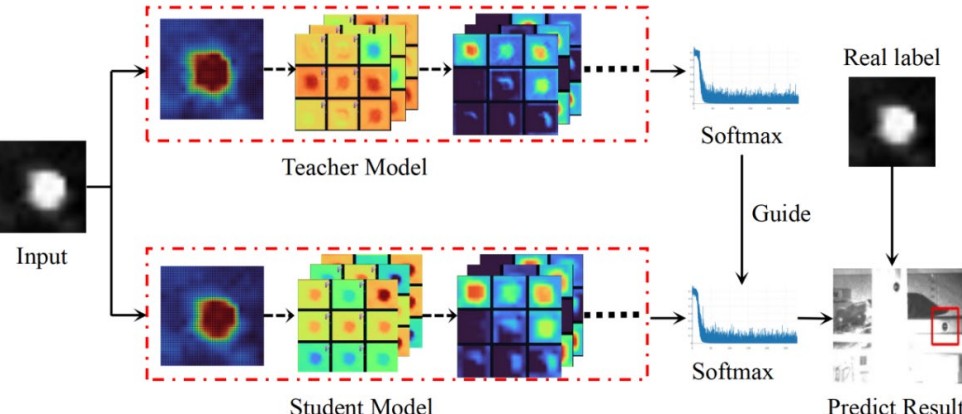

**Figure 2.** Knowledge distillation schematic diagram.

The temperature coefficient $T$ is added to the output layer of Softmax to smooth the probability distribution of the network's output. The output obtained is called a soft target; soft targets and real tags work together to guide student network training. The loss function $J_{KD}$ can be expressed as

$$J_{KD} = J_{CE}(y_{true}, p) + \gamma T^2 J_{CE}\left(\hat{p}_i, \hat{q}_i\right) \tag{1}$$

$$\hat{q}_i = \frac{\exp(z_i/T)}{\sum_j \exp(z_i/T)} \tag{2}$$

Neural networks typically generate class probabilities using a Softmax output layer, which normalizes $z_i$ to probability $q_i$. Furthermore, $J_{CE}(y_{true}, p)$ represents the cross-entropy between the predicted output of the student network and the true label, and $\gamma$ is a hyperparameter that adjusts the proportion between the predicted output after smoothing by the student network and the teacher network. When cross-entropy is backpropagated, the gradient changes to the original $1/T^2$, which is smoothed by a hyperparameter $T$. Therefore, to preserve the scale of its gradient consistent with the scale of the cross-entropy corresponding to the true label, it is necessary to multiply the smoothed cross-entropy by $T^2$.

### 3.2. TensorRT Acceleration

TensorRT can reconstruct the computational graph according to the structural characteristics of different deep learning models and then generate engine files that are more suitable for inference operations. It will select the most suitable CUDNN [26] for acceleration operations, significantly reducing the inference time of deep learning models. The core part of TensorRT is layer and tensor fusion. It is well known that the excellent performance of deep learning is due to its numerous network layers (convolution layer, batch-norm layer [27], activation layer, etc.). For regular model deployment inference, the GPU performs layer-by-layer computation. GPU has a shorter computation time than CPU because it exploits different CUDA cores. During this operation, a lot of time will be concentrated on CUDA memory occupation to read and write the weights of each layer, leading to a serious waste of GPU resources. To alleviate this phenomenon, TensorRT will perform transverse merging between layers and longitudinal merging between layers.

In this experiment, we selected the lightweight network Shuffv2_x0_5 to detect photoelectric targets (the specific reasons for selecting the network are analyzed in the experimental part of Section 4.1). The Shuffv2_x0_5 network ONNX [28] and engine computational diagram are shown in Figure 3:

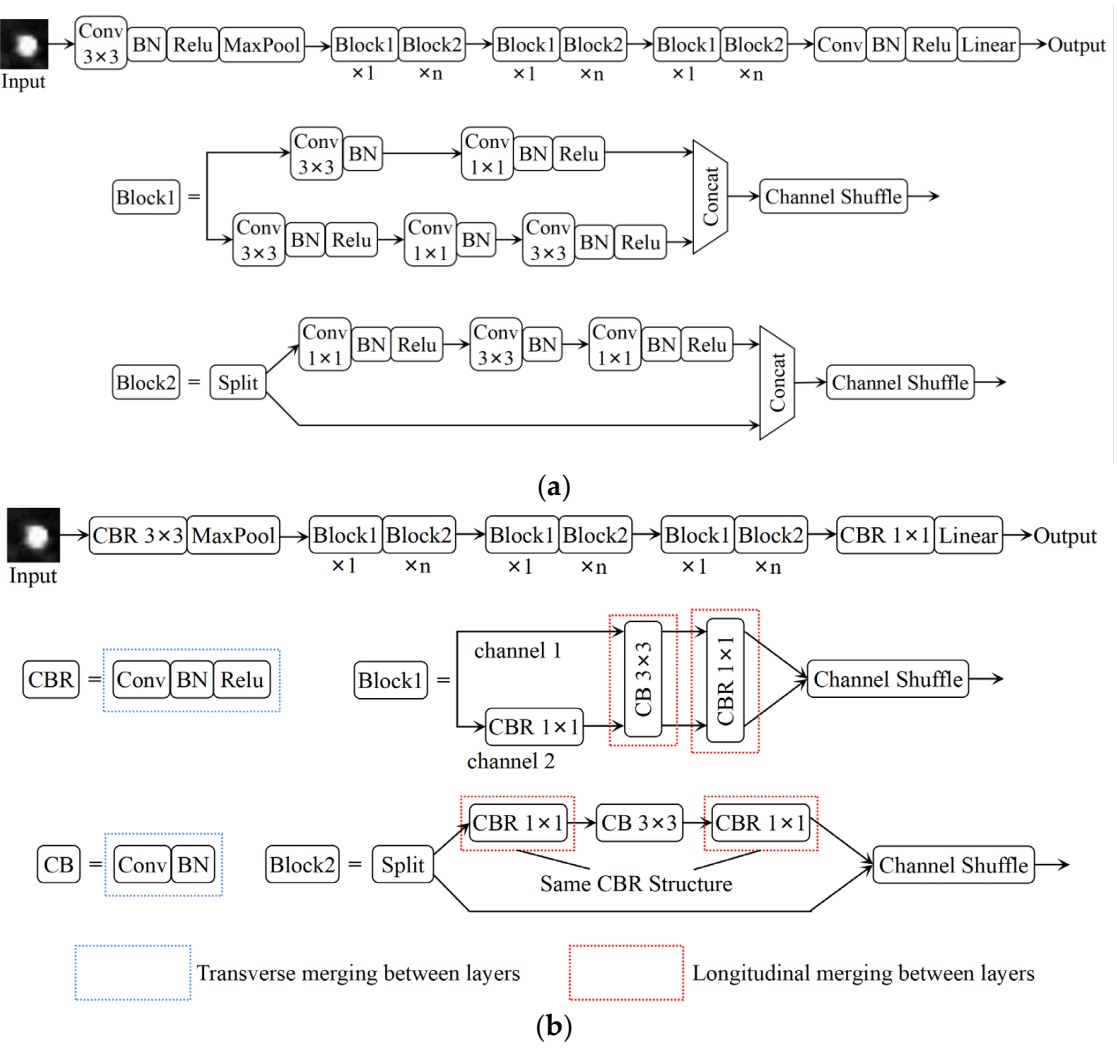

**Figure 3.** ShuffleNetV2 network structure: (**a**) ONNX format file calculation diagram and (**b**) engine format file calculation diagram.

In the process of deploying model inference, the operation of each layer is completed by the GPU by booting different CUDA core calculations. Although the CUDA core computing tensor is very fast, much time is wasted to start up CUDA cores to read and write tensor on each layer. This causes a bottleneck in memory bandwidth and a serious waste of GPU resources. For example, in the Figure 3 Block 1 structure, Conv 1 × 1 layer, Conv 3 × 3 layer, *BN* layer, and R*elu* layer each need to start different CUDA cores. The vast majority of computation in neural networks is concentrated in the Conv layer and the R*elu* layer. TensorRT will conduct transverse merging between layers. As shown in Figure 3b, the Conv layer and *BN* layer are transverse merged into a CB structure. Transverse merges the Conv layer, *BN* layer, and R*elu* layer into a CBR structure. This strategy reduces the memory footprint by occupying only one CUDA core. The Conv layer can be expressed as:

$$Y_{conv} = \sum_i^N w_i x_i + b \tag{3}$$

where $x_i$ is the input tensor of the layer's shape, $C \times H \times W$, $Y_{conv}$ is the output tensor of the layer's shape, $C \times H \times W$, $w_i$ is the weight parameter, and $b$ is the bias parameter. The *BN* layer can be expressed as

$$Y_{BN} = \frac{X_{BN} - mean_{BN}}{\sqrt{var_{BN} + epsilon}} \cdot scale + \beta \tag{4}$$

where $mean_{BN}$ is the mean of the input tensor, $var_{BN}$ is the variance of the input tensor, and $\beta$ is a learnable parameter. In TensorRT optimization, the Conv layer is usually merged with the *BN* layer to reduce the memory footprint. Take the output $Y_{conv}$ of the Conv layer in Equation (1) as input $X_{BN}$ of the *BN* layer. After the Conv layer is fused with the *BN* layer, the merged layer can be expressed as:

$$Y_{BN} = \frac{\sum_i^N w_i \cdot scale}{\sqrt{\mathrm{var}_{BN} + epsilon}} \cdot x_i + \frac{b - mean_{BN}}{\sqrt{\mathrm{var}_{BN} + epsilon}} \cdot scale + B_{BN} \tag{5}$$

where the weight parameter *W* and the paranoid parameter *B* after fusion can be expressed as:

$$W = \frac{\sum_i^N w_i \cdot scale}{\sqrt{\mathrm{var}_{BN} + epsilon}} \tag{6}$$

$$B = \frac{b - mean_{BN}}{\sqrt{\mathrm{var}_{BN} + epsilon}} \cdot scale + \beta \tag{7}$$

During the float32 or float16 quantification, the Re*lu* layer can be fused with the Conv + *BN* layer. For the input vector from the previous convolutional layer, the output can be obtained using the linear rectification activation function. The fusion result can be expressed as:

$$Y_{\mathrm{Re}lu} = \max(0, Y_{BN}) \tag{8}$$

Furthermore, the Shuffv2_x0_5 network structure in Figure 3a reduces the 244 layers to 135 layers through the transverse merging of TensorRT. Moreover, TensorRT further conducts longitudinal merging between layers by merging layers of the same structures and different values, which also occupies only a single CUDA core. For example, in Figure 3 Block 1, the CBR contains Conv 1 ×1 layers, *BN* layers, and Re*lu* layers of channel 1 and channel 2. The Shuffv2_x0_5 network is reduced to 106 layers through longitudinal merging between the layers. The merged CBR structure will traverse CUDA depthwise convolution, fused contact convolution, CuDNN convolution, CuBLAS convolution, cask convolution, and other tactics, and then choose the fastest tactic for the operation. After TensorRT optimization, the number of computational layers significantly reduces. Meanwhile, the CUDA cores and memory also reduce to achieve inference acceleration. Overall, the TensorRT quantification tool provides a complete automated calibration process in addition to the transverse and longitudinal merging between layers. It avoids the complicated and cumbersome workload of manual parameter adjustment, and its quantification effect is superior.

### 3.3. Multi-Stream Mode

The optimization of general TensorRT mainly includes the multi-optimization profile mode, multi-context mode, and multi-stream mode. Among them, the multi-optimization profile mode solves the problem that TensorRT needs to be compatible with multiple different dynamic sizes when performing optimization on kernels, and the kernel performs poorly for some specific sizes when the dynamic size is relatively large. The multi-context mode solves the problem that the engine saves duplicates during the TensorRT multi-threaded inference. It can be achieved by saving only one engine file on the GPU for multiple threads to perform inference calculations, and the memory footprint and inference computation can be superimposed simultaneously. The multi-stream mode solves the problem of excessive time wasted due to data copying and overlapping calculations during engineering applications. Essentially, the multi-stream mode increases GPU usage and increases inference speed.

According to the actual real-time detection requirements for detecting optoelectronic targets, the input data dimensions are fixed values when inference is performed in Shuffv2_x0_5 models. Therefore, we only use the multi-stream mode to further optimize the general TensorRT inference in CUDA computing GPU-related function calls. For exam-

ple, asynchronous memory request/release, asynchronous memory copy, kernel execution, and other operations will be placed in the stream. In the same stream, the order of function operation is gradually operated by the time order in which each step joins the stream. When performing tasks of different streams, we must insert CUDA events to control stream timing synchronization. As shown in Figure 4:

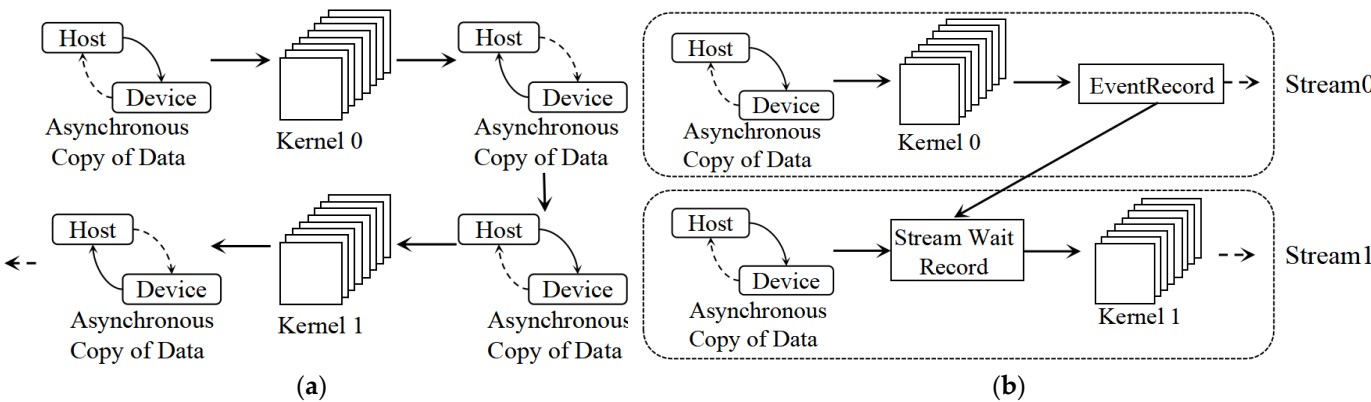

**Figure 4.** Stream inference mode execution order: (**a**) single-stream, (**b**) multi-stream.

In multi-stream mode, Stream 0 is executed sequentially (Asynchronous copy of data → Kernel 0 → EventRecord → Asynchronous copy of data). Stream 1 is executed sequentially (Asynchronous copy of data → Stream Wait Record → Kernel 1 → Asynchronous copy of data). The difference is that Stream 1 waits when it executes Stream Wait Record. Moreover, Stream 1 will continue to execute when Stream 0 finishes executing the EventRecord step. This ensures that Kernel 1 is executed only after Kernel 0 is executed, which controls the order in which different kernels are executed between different streams. It is worth mentioning that the CPU-side data of all CUDA event function calls is returned immediately. The memory in the CPU side is divisible that swaps data into a file in order to conserve memory usage. This will lead to an undesirable phenomenon. The memory required by the CPU has been swapped to a file when the kernel is not executed. To solve this problem, we need to use pinned-memory in the multi-stream mode to call the CUDA stream function and the CUDA event function. This saves resources in time and space; multi-streams transfer each other between asynchronous copies and accelerated inference calculations.

## 4. Results

### 4.1. Analysis of Knowledge Distillation Results

In this section, we conducted relevant experiments in order to verify that knowledge distillation can eliminate the poor real-time performance of convolutional neural networks deployed in GPU computing frameworks with limited computing power, and to improve the prediction accuracy of lightweight network models. The hardware device used in the experiment is the NVIDIA Jeston Nano, well-received industry hardware that is commonly used as a cost-effective edge device. The knowledge distillation teacher network mainly selects three classical neural networks: VGG16, AlexNet, and Resnet18.

The number of parameters and calculations are two important indicators to measure the deep learning method. The top1%, parameter amount, calculation amount, and average inference time of 50 inferences in the NVIDIA Jeston Nano are reported in Table 1 (—indicates that the resources occupied by NVIDIA Jeston Nano are too large to run).

**Table 1.** Candidate teacher network Top-1%, parameter quantity, calculation amount, and inference time in NVIDIA Jeston Nano.

| Network | Top-1% | Parameter Quantity (M) | Calculated Amount (M) | Inference Time (s) |
|---------|--------|------------------------|-----------------------|--------------------|
| VGG16 | 99.18% | 134.27 | 1368.74 | - |
| AlexNet | 98.75% | 57.00 | 90.61 | 0.2152 |
| ResNet18 | 98.27% | 23.51 | 82.27 | 0.1343 |

Table 1 shows that the three classical neural networks VGG16, AlexNet, and Resnet18 have little effect on the detection accuracy of photoelectric targets. The difference lies in the complexity of the respective network structure. It leads to large differences in the number of parameters and calculations of the network. It will affect the inference speed of the network.

Therefore, we carried out knowledge distillation comparison experiments between teacher networks and different student networks. The student network was one of the lightweight networks: Shuffv2 [29], Squeezent [30], GhostNet [31], and CondenseNetv2 [32]. The parameters of top-1%, parameter quantity, calculation amount, and average inference time of 50 inferences in the NVIDIA Jeston Nano after knowledge distillation on the four lightweight networks are reported in Table 2:

**Table 2.** Lightweight network top-1% before and after knowledge distillation, parameter quantity, calculation amount, and the inference time in NVIDIA Jeston Nano.

| Network | Top-1% | KD-VGG16 Top-1% | KD-AlexNet Top-1% | KD-ResNet18 Top-1% | Parameter Quantity (M) | Calculated Amount (M) | Inference Time (s) |
|---------|--------|-----------------|-------------------|--------------------|------------------------|-----------------------|--------------------|
| Shuffv2_x0_5 | 97.31% | 96.47% | 96.08% | 97.72% | 0.34 | 2.95 | 0.0759 |
| Shuffv2_x1_0 | 97.87% | 97.22% | 97.74% | 98.09% | 1.26 | 11.62 | 0.0893 |
| Shuffv2_x1_5 | 98.23% | 97.37% | 97.89% | 98.18% | 2.48 | 24.07 | 0.0982 |
| Shuffv2_x2_0 | 98.39% | 97.93% | 98.14% | 98.21% | 5.35 | 47.62 | 0.1176 |
| Squeezent1_0 | 96.06% | 86.83% | 66.72% | 95.97% | 0.73 | 41.74 | 0.0734 |
| Squeezent1_1 | 90.04% | 93.47% | 76.79% | 92.29% | 0.72 | 16.05 | 0.0697 |
| GhostNet | 96.22% | 97.72% | 97.60% | 97.67% | 3.90 | 14.26 | 0.0928 |
| CondenseNetv2 | 93.87% | 96.94% | 95.02% | 97.23% | 7.26 | 169.0 | - |

Table 2 reveals that most of the lightweight networks by knowledge distillation have higher detection accuracy than the lightweight networks trained alone. However, this is not absolute. According to the Kolmogorov complexity, reducing the complexity of the dataset can improve the accuracy of machine learning models [33,34]. So model accuracy is determined by the complexity of the dataset and the training process. In particular, the accuracy of the ResNet18 network for knowledge distillation is more significant. The result is that the ResNet18 network is less different from the lightweight network, which is convenient for obtaining the feature information passed between layers. The lightweight network can obtain better distillation results. Therefore, the ResNet18 was selected as the teacher network for knowledge distillation. In the selection of student networks, inference time is the primary factor considering that the algorithm is to be deployed on a low-computing GPU computing framework. Table 2 shows that the Squeezent1_1, Squeezent1_0, and Shuffv2_x0_5 after knowledge distillation have good reasoning performance. Secondly, we must consider the detection accuracy, parameter quantity, and calculation amount factors. It can be found that Shuffv2_x0_5 has more advantages in these three parameters through experimental comparison. Shuffv2_x0_5 accuracy increased by 1.75% compared to Squeezent1_0 and 5.43% compared to Squeezent1_1. This analysis is because Shuffv2_x0_5 uses a residual network structure similar to ResNet18. In layperson's terms, the main module of the Shuffv2_x0_5 is the improved and lightweight design in the module of

ResNet18. At the same time, Shuffv2_x0_5 has a significant reduction in the number of model parameters and calculations, which has better performance advantages. Therefore, the Shuffv2_x0_5 network after knowledge distillation of ResNet18 was selected and deployed as a photoelectric target detection network on the NVIDIA Jeston Nano. During the training process, Shuffv2_x0_5 loss and accuracy after individual training and ResNet18 knowledge distillation are shown in Figure 5:

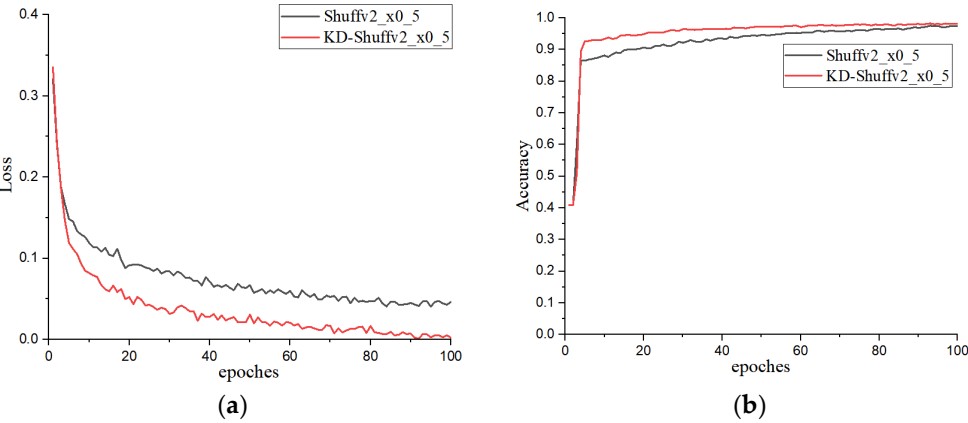

(a)　　　　　　　　(b)

**Figure 5.** Knowledge distillation contrasts loss and accuracy in the process of model training: (**a**) loss comparison chart, (**b**) accuracy comparison chart.

*4.2. Analysis of MSIOT Result*

In this section, we conducted experiments and analyzed the effectiveness of the MSIOT method in GPU computing frameworks with limited computing power. We used the TensorRT engine for inference acceleration and analysis of its inference results. The optimal lightweight network is Shuffv2_x0_5 (hereinafter referred to as ShuffNet) according to experiment 4.1. We converted ShuffNet to a compatible ONNX file and completed the construction of the engine file (hereinafter referred to as ShuffEng). Since the NVIDIA Jetson Nano platform only supports F32 precision, the engine adopts the full precision mode for inference. Figure 6 illustrates a comparative test of the photoelectric target detection forward inference between ShuffNet and ShuffEng by CUDNN in GPU mode on the NVIDIA Jetson Nano, and the true classified target is marked on the original image.

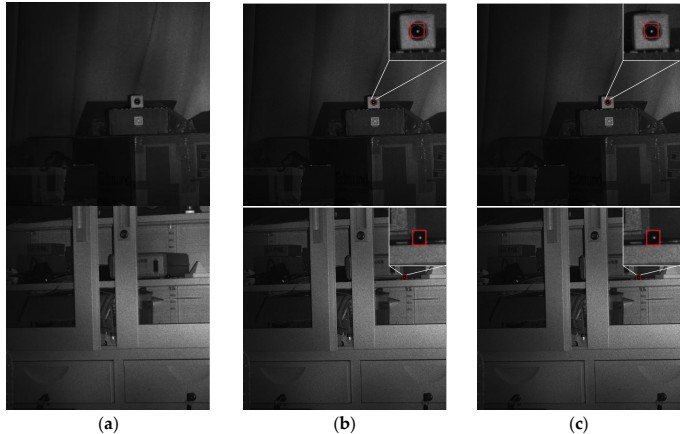

(a)　　　　　　(b)　　　　　　(c)

**Figure 6.** Photoelectric target detection forward inference comparison test: (**a**) active image, (**b**) inference result of ShuffNet, and (**c**) inference result of ShuffEng.

We input 500 test sets of experiment 2.2 into ShuNet and ShuEng, respectively, for photoelectric target binary classification tasks, and the confusion matrix is identical, as shown in Figure 7:

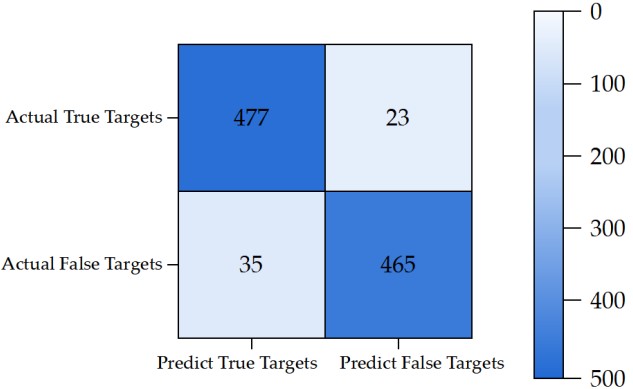

**Figure 7.** Part of the test dataset photoelectric target binary classification confusion matrix.

Accuracy, precision, recall, and F1 score have always been essential detection indicators for binary classification tasks. The above four parameters are used to evaluate the detection results of ShuNet and ShuEng according to the confusion matrix, as shown in Table 3.

**Table 3.** Detection rate and false alarm rate of different inference modes in multiple scenarios (unit: %).

| Model | Accuracy | Precision | Recall | F1 Score |
|---|---|---|---|---|
| ShuffNet | 94.2% | 93.16% | 95.4% | 94.27% |
| ShuffEng | 94.2% | 93.16% | 95.4% | 94.27% |

ShuffNet inference weight precision is 32 bits. ShuffEng inference full precision is also 32 bits. It is fully verified through Table 3 that ShuffNet and ShuffEng have no impact on the accuracy, precision, recall, and F1 score indicators of photoelectric target detection when the inference weight type is the same and only affects the inference speed of the model. Next, we explored the inference time of ShuffNet and ShuffEng on the NVIDIA Jeston Nano hardware platform. We selected 500 test sets of different specifications for inference, and the average inference time of each 10 times was recorded as data. The inference times for different sizes ($20 \times 20$, $40 \times 40$, $60 \times 60$, $80 \times 80$, $100 \times 100$, $120 \times 120$) input to ShuffNet and ShuffEng are shown in Figure 8:

The speedup ratio of different sizes ($20 \times 20$, $40 \times 40$, $60 \times 60$, $80 \times 80$, $100 \times 100$, $120 \times 120$) input to ShuffNet and ShuffEng is shown in Table 4:

**Table 4.** ShuffEng inference time speedup ratio for different input sizes.

| Input Size | $20 \times 20$ | $40 \times 40$ | $60 \times 60$ | $80 \times 80$ | $100 \times 100$ | $120 \times 120$ |
|---|---|---|---|---|---|---|
| Speedup Ratio | 7.2323 | 6.1883 | 6.9185 | 5.7292 | 6.2322 | 4.3423 |

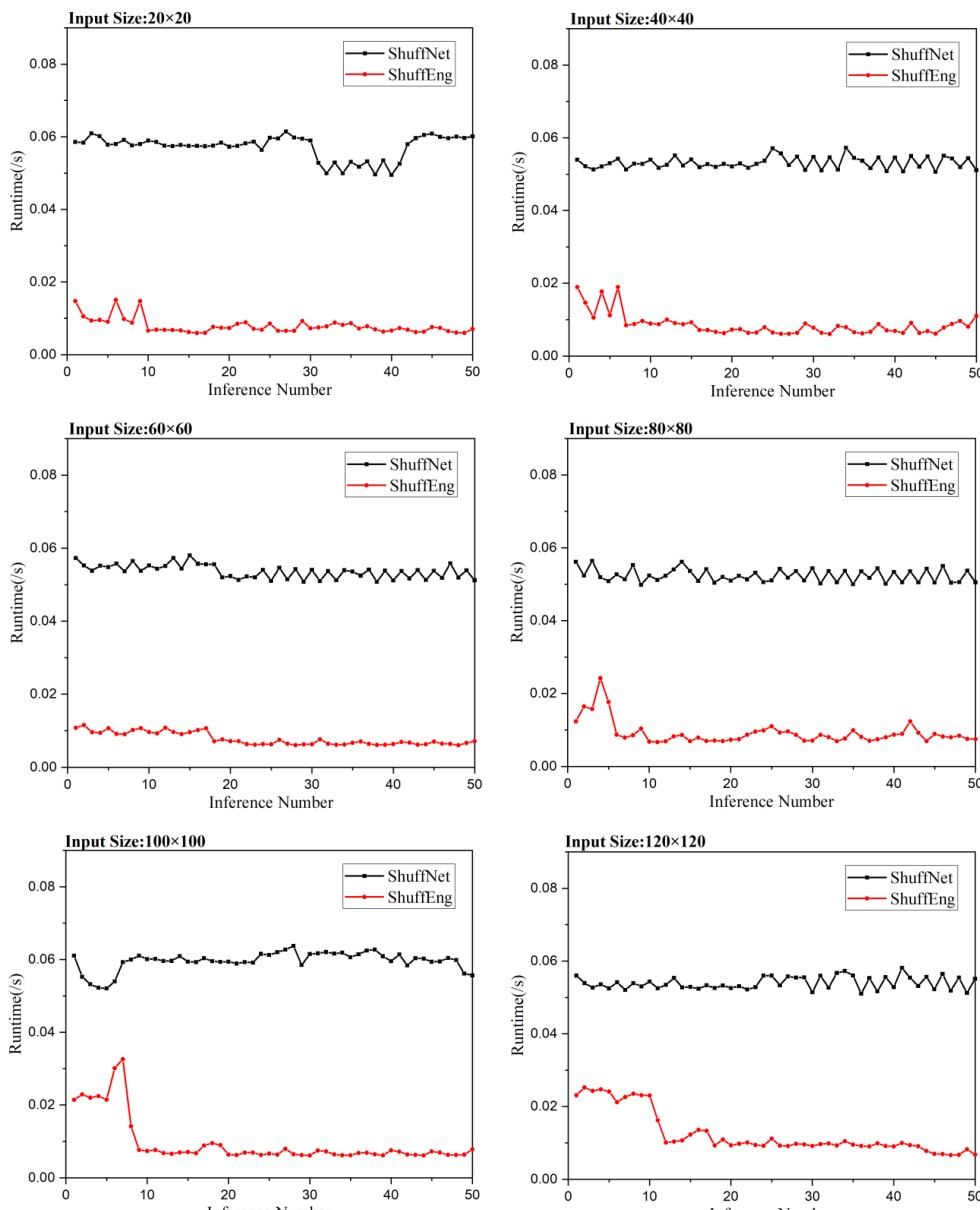

**Figure 8.** Inference time input to ShuffNet and ShuffEng with different sizes.

Figure 8 and Table 4 demonstrate a significant increase in ShuffEng inference time compared to ShuffNet. The increase in the input size increases the inference time of the model. According to different input sizes, the inference time acceleration ratio can reach 4.3–7.2 ×, significantly improving the inference performance of deep learning models deployed in low-computing GPU computing frameworks. However, in the real-time detection of photoelectric targets, the number of detection areas of different frame images differs due to the possibility of multiple targets in the same image.

Multiple detection regions in the image are input into the network model simultaneously. The difference in the number of detection regions will lead to a significant difference in the inference time between the nth and the n–1 frame images. It can occur because of excessive waiting time caused by the data copy. To solve this problem, we used multi-stream to tuning TensorRT and conduct comparative experiments. We selected 500 test sets for inference time comparison during the experiment and recorded the average of every 10 inference times as one dataset to further narrow the error of single inference time. We recorded host to device data copy (Copy-HtoD), device to host data copy (Copy-DtoH),

TensorRT runtime, and MSIOT (Ours) runtime. The inference time of TensorRT and MSIOT during photoelectric targets real-time detection is shown in Figure 9.

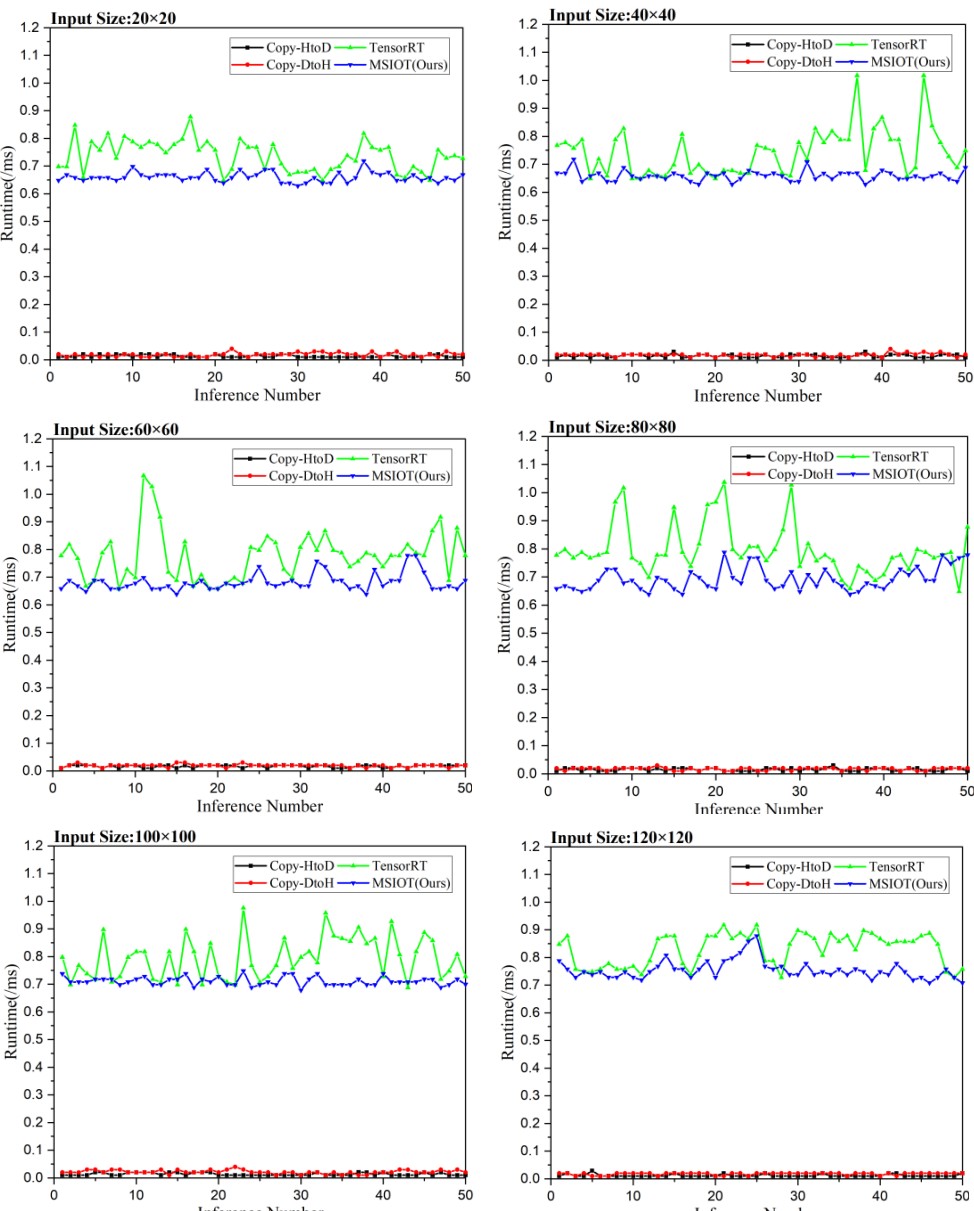

**Figure 9.** Inference time comparison between TensorRT and MSIOT during photoelectric targets real-time detection.

Figure 9 shows that MSIOT has better inference performance. MSIOT is 9.3% faster than TensorRT, and the inference time is more stable. The reasons for the improvement of inference speed are mainly reflected in the following two aspects:

1. Copy-HtoD and Copy-DtoH cannot be avoided when using GPUs for inference. Figure 7 shows that Copy-HtoD and Copy-DtoH occupy a relatively small amount of time, but the time occupancy of data replication in each inference will have a non-negligible time cost. Theoretically, MSIOT saves n–1 times Copy-HtoD and Copy-DtoH in the n times inference process compared to TensorRT. MSIOT can better cope with multiple inference tasks of photoelectric targets real-time detection.
2. MSIOT uses multiple threads within the same CUDA to perform calculations simultaneously by calling CUDA cores in a more complete way than TensorRT, improving the

hardware performance utilization and stabilizing the inference speed in performing photoelectric targets detection task.

This paper proposes a MSIOT inference acceleration method. It improves the inference speed of photoelectric target detection without affecting accuracy. Its single inference time is stable between 0.6–0.8 ms. This method has practical engineering application value.

### 4.3. System Experimental Verification

In order to verify the effectiveness and robustness of the above method in actual indoor scenarios, this chapter integrates a photoelectric target rapid detection system and carries out experimental system verification. We pre-placed 1–2 pinhole cameras in different indoor scenarios. We hid a pinhole camera in the corner of a complex interior or in an object that is difficult to identify by the human eye alone. The following experiments were carried out for the following four scenarios, with distances ranging from 3–10 m, as shown in Figure 10:

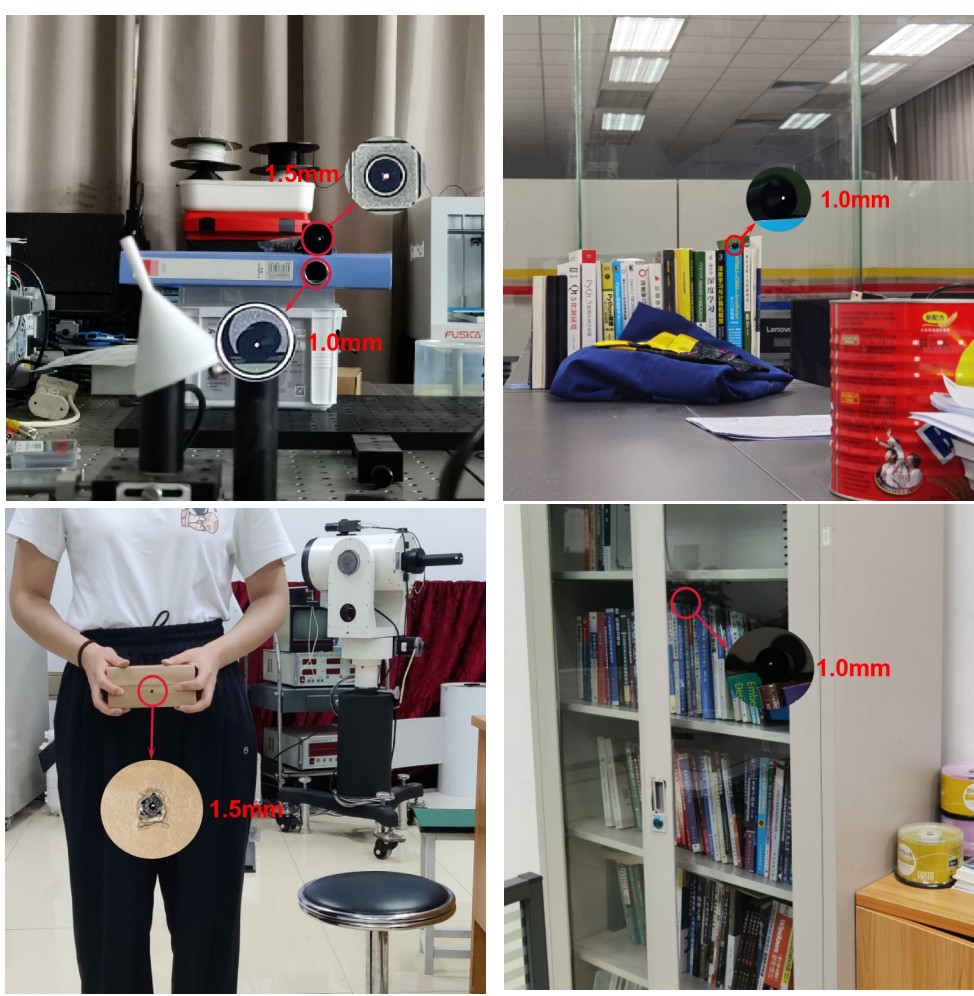

**Figure 10.** Four different indoor experimental scenarios.

Scenario 1 is a laboratory scenario where two pinhole cameras are hidden; Scenario 2 is a conference room scenario hiding a 1.0 mm diameter pinhole camera on a book; Scenario 3 simulates a 1.5 mm diameter pinhole camera installed in a handbag to shoot the field of view ahead; Scenario 4 is a scenario where a 1.0 mm diameter pinhole camera is hidden in a bookcase with human interference factors. The above four scenarios are challenging to find by naked eye inspection alone, and the detection results of the photoelectric target rapid detection system are shown in Figure 11:

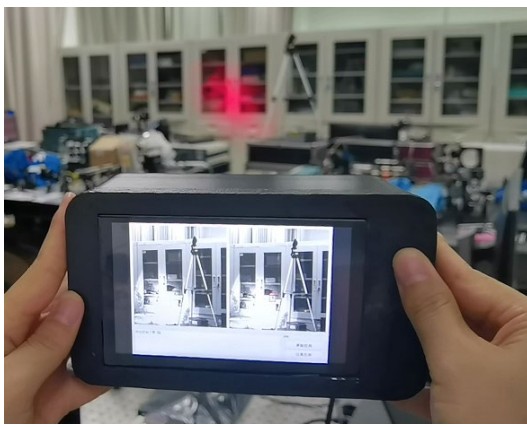

**Figure 11.** The system detects the image.

The real-time detection results of the equipment are shown in Figure 12, and they are all real-time detection.

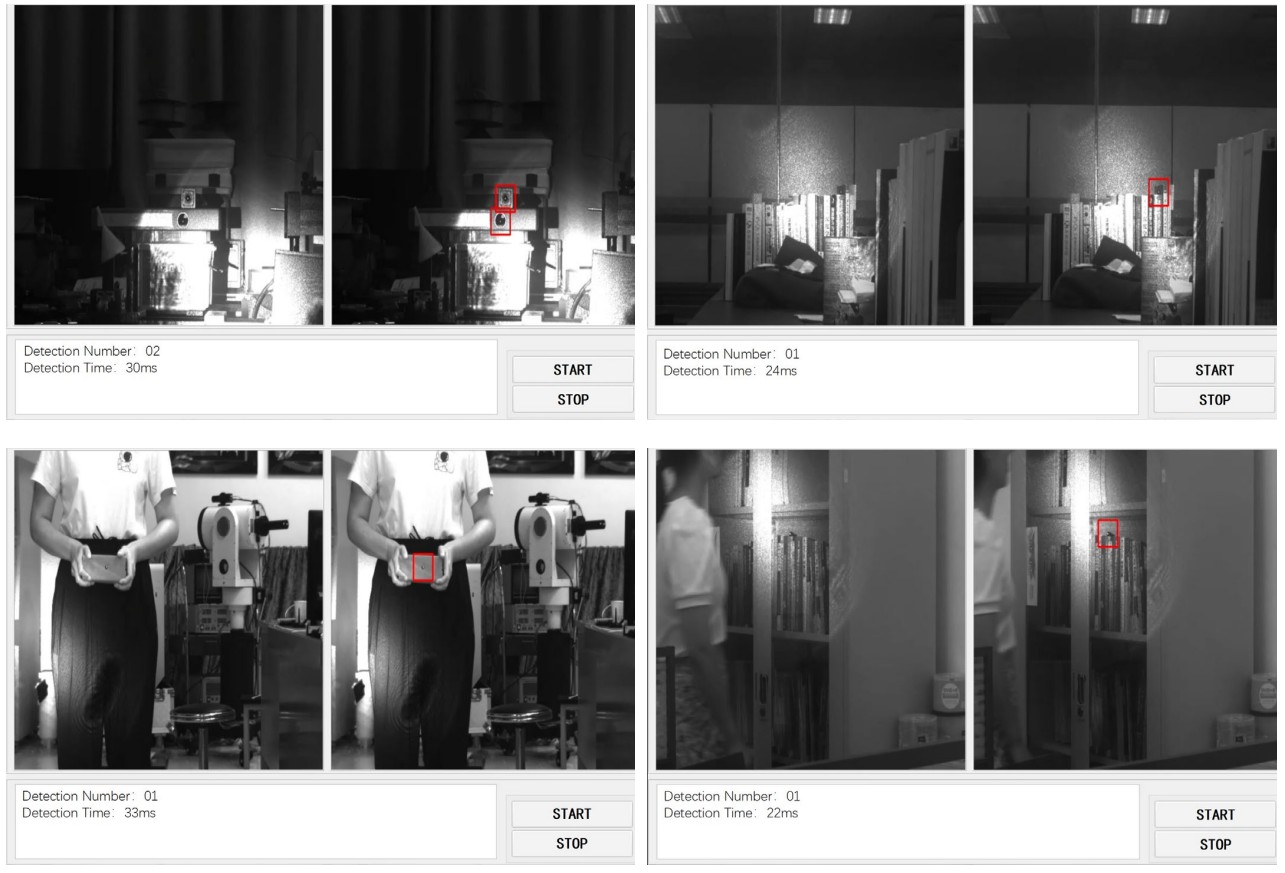

**Figure 12.** The system detects the results in real time.

The equipment can accurately detect pinhole cameras in different indoor environments through the above real-time detection results. It can still detect in real time with reasonable accuracy under human interference factors, proving the equipment's effectiveness and the superiority of surpassing manual troubleshooting.

While verifying the system effectiveness of the above experiments, the detection time of each scenario was counted, and the average detection time of the four scenarios was 30 ms, 24 ms, 33 ms, and 22 ms, respectively.

Scenario 3 is more complex than the Scenario 2 background. The background in Scenario 2 is only miscellaneous objects and some books. The background of Scenario 3 contains many bright metal objects, such as instruments, stools, and connecting wires. These items have specific pseudo-target characteristics. After analysis, the length of detection time depends on the number of candidate regions input. The more bright radiation in the background, the more complex the background, resulting in a sharp increase in candidate regions so that the discrimination time becomes longer.

We counted the detection time in 10 different indoor scenarios, and according to the 1-s video composed of 24 frames, each detection time's average detection time is less than 42 ms during the actual detection process to ensure the real-time detection video. After statistics, the average detection time of the system for ten different indoor scenarios is 32 ms, and the performance of similar methods is compared, as shown in Table 5.

**Table 5.** Comparison of TP, recall, and average time with different methods.

| Method | TP | Recall | Average Time |
|---|---|---|---|
| Ke [15] | - | 75.5% | 5.36 s |
| Liu [15] | 466 (500) | 93.2% | 166 ms |
| Huang [15] | 189 (200) | 94.5% | 104 ms |
| MSIOT(Ours) | 477(500) | 95.4% | 32 ms |

From Table 5, it can be concluded that MSIOT is better than the method proposed by Ke [15] and Liu [15] in terms of recall, which is the same as the YOLOv3-4L method proposed by Huang [15]. The MSIOT model compression and acceleration are much shorter in detection time than the other three methods. MSIOT can prove the system detection function's superior processing speed and real-time performance.

## 5. Conclusions

We propose a MSIOT photoelectric target detection inference acceleration method, which solves the problem of slow running time and poor real-time performance of deep learning network models deployed in low-computing power GPU computing frameworks. MSIOT is optimized through multi-stream by combining knowledge distillation and TensorRT accelerated reasoning. Extensive experimental results demonstrate that MSIOT effectively improves the prediction accuracy of the lightweight network model, and the inference speed of the model can be further improved on the basis of TensorRT. The entire process effectively reduces the computing power requirements of neural networks deployed in GPU computing frameworks, which is significant for designing a real-time photoelectric target detection system. The disadvantage of the method is that the hardware part is not accelerated enough and can be further improved in the future according to the hardware characteristics.

**Author Contributions:** Conceptualization, Y.Z. and H.G.; methodology, L.Z.; software, S.Z.; validation, L.Z. and S.Z.; formal analysis, S.Z.; investigation, S.Z.; resources, L.Z.; data curation, L.Z., S.M. and Y.C.; writing—original draft preparation, S.Z.; writing—review and editing, S.Z. and L.Z. All authors have read and agreed to the published version of the manuscript.

**Funding:** This research received no external funding.

**Data Availability Statement:** Due to privacy reasons, datasets are not publicly available.

**Conflicts of Interest:** The authors declare no conflict of interest.

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
