# Peer review of "Inference-Optimized High-Performance Photoelectric Target Detection Based on GPU Framework"

_photonics, doi:10.3390/photonics10040459_

Round 1
Reviewer 1 Report
Please read and “fully” address the comments listed below:
1. The ABSTRACT is not written in a logical order (it is a bit lengthy). Start with an overview of the topic and a rationale for your paper. Describe the methodology you used and the general outline of the manuscript. Also, state the result in more detail (i.e., provide some numbers). Finally, detail the implications of your research (and future directions).
2. The novelty of your work is still unclear to the reader, which should be further detailed both in the Abstract and Introduction. In other words, the purpose of the research is missing, which must be clearly presented.
3. It is mentioned that the method’s accuracy rate is 94.20%, is this value ideal and conforms well to industry standards? In addition to “accuracy” could you provide “precision” and “recall” indices?
4. Provide accuracy vs epoch “graphs” of train and test sets to determine the model’s accuracy and performance.
5. Provide more explanation for this sentence: Page 17, Line 211: “However, different streams need to insert CUDA events to control the execution 211 order between the Multi-Stream”.
6. Similarly, this sentence needs to be better explained: Page 11, Line 315: “However, there will be multiple regions entering at the same time, and the time difference between the previous inference and the next inference is large”.
7. Determine how the hyperparameters of your deep learning model were optimized.
8. The authors mentioned that “most of the lightweight networks by knowledge distillation have higher detection accuracy than the lightweight networks trained alone.”. However, this statement is partially true because the dataset might be highly complex such that even a better network may not necessarily improve the model accuracy. In machine learning, this can refer to “Kolmogorov complexity” denoting the length of the shortest computer program that produces the output. Therefore, write a paragraph in your paper arguing that reducing the complexity of your dataset can potentially improve the accuracy of the deep learning model and reference the 2 papers listed below (that reduce the complexity of their dataset to improve the accuracy of their machine learning models)
· Bolon-Canedo, V., & Remeseiro, B. (2020). Feature selection in image analysis: a survey. Artificial Intelligence Review, 53(4), 2905-2931.
· Kabir, H., & Garg, N. (2023). Machine learning enabled orthogonal camera goniometry for accurate and robust contact angle measurements. Scientific Reports, 13(1), 1497.
10. Conclusion: Can authors highlight future research directions and recommendations? Also, highlight the assumptions and limitations (e.g., shortcomings of the present study). Besides, recheck your manuscript and polish it for grammatical mistakes (you can use “Grammarly” or similar software to quickly edit your document).
11. How can the results of this study be applied to other fields beyond photoelectric target detection, and what are some of the potential implications for future research in this area?
12. What are some of the challenges faced when deploying deep learning models in GPU computing frameworks with limited computing power?
Author Response
The reviewers are thanked for their suggestions, and the professionalism of the reviewers is seen through the suggestions. The following answers are given to the questions raised by the reviewers, and mark the article in yellow.
Q1: The ABSTRACT is not written in a logical order (it is a bit lengthy). Start with an overview of the topic and a rationale for your paper. Describe the methodology you used and the general outline of the manuscript. Also, state the result in more detail (i.e., provide some numbers). Finally, detail the implications of your research (and future directions).
A1: Thank you to the reviewers for their suggestions. I rewrote the abstract section to add sections such as study background, method description, result description, and research significance, as follows:
Deep learning has better detection efficiency than typical methods in photoelectric target detection. However, classical CNNs on GPU frameworks consume too much computing power and memory resources. We propose a Multi-Stream inference-optimized TensorRT (MSIOT) method to solve this problem effectively. MSIOT uses knowledge distillation to effectively reduce the number of model parameters by layer guidance between CNNs and lightweight networks. Moreover, we cooperate with TensorRT and Multi-Stream mode to reduce the number of model computation. MSIOT again increases inference speed by 9.3% based on the 4.3-7.2x acceleration of TensorRT. The experimental results show that the model's mean average accuracy, precision, recall, and F1 score after distillation can reach up to 94.20%, 93.16%, 95.4%, and 94.27%. It is of great significance for designing a real-time photoelectric target detection system.
Q2: The novelty of your work is still unclear to the reader, which should be further detailed both in the Abstract and Introduction. In other words, the purpose of the research is missing, which must be clearly presented.
A2: Thank you to the reviewers for their suggestions. In order to provide readers with a better reading experience, a description of the novelty of the work has been added to the abstract and introduction sections, as follows:
Classical CNNs running on GPU compute frameworks [18-20] create problems with poor real-time performance. Therefore, this paper proposes a Multi-Stream inference-optimized TensorRT (MSIOT) inference acceleration method. This method verifies the real-time detection of photoelectric targets on the GPU computing frame with limited computing power. This paper’s significant contributions are as follows:
- In order to effectively reduce the number of model parameters and ensure the high accuracy of the neural network. We train a high-precision CNN model through knowledge distillation [24]; guided learning is performed on lightweight networks. Finally, a high-precision lightweight network model is obtained.
- In order to reduce the number of computation in the process of model inference, we deeply explore the inference acceleration principle of the TensorRT [21-23] engine based on the characteristics of the GPU computing framework and build a computational graph based on the existing network. Experiments verify the effectiveness and practicality of TensorRT inference acceleration.
- In order to solve the excessive waste of time for data replication and overlapping calculations during model inference. We optimize TensorRT to exploit CUDA (Computer Unified Device Architecture) control based on the kernel execution principle. The utilization of the GPU is fully invoked through the Multi-Stream [25] mode, further shorting the inference time of deep learning models.
Q3: It is mentioned that the method’s accuracy rate is 94.20%, is this value ideal and conforms well to industry standards? In addition to “accuracy” could you provide “precision” and “recall” indices?
A3: Thank you to the reviewers for their questions. This value may have some idealization, and there are many influencing factors such as laser power, natural light intensity, atmospheric turbulence, background complexity and so on in the process of photoelectric target detection. We only test for the environment in which we are currently in. This article adds accuracy, precision, recall, F1 score indicators in Section 4.2, as follows:
We input 500 test sets of experiment 2.2 into ShuNet and ShuEng respectively for photoelectric target binary classification tasks, and the confusion matrix is identical, as shown in Figure 7:
Figure 7. Part of the test dataset photoelectric target binary classification confusion matrix.
Accuracy, Precision, Recall, and F1 score have always been essential detection indicators for binary classification tasks. The above four parameters are used to evaluate the detection results of ShuNet and ShuEng according to the confusion matrix, as shown in Table 3.
Table 3. Detection rate and false alarm rate of different inference modes in multiple scenarios (unit: %)
|
Model |
Accuracy |
Precision |
Recall |
F1 score |
|
ShuffNet |
94.2% |
93.16% |
95.4% |
94.27% |
|
ShuffEng |
94.2% |
93.16% |
95.4% |
94.27% |
ShuffNet inference weight precision is 32 bits. ShuffEng inference full-time precision is also 32 bits. It is fully verified through Table 3 that ShuffNet and ShuffEng have no impact on the Accuracy, Precision, Recall, and F1 score indicators of photoelectric target detection when the inference weight type is the same and only affect the inference speed of the model.
Q4: Provide accuracy vs epoch “graphs” of train and test sets to determine the model’s accuracy and performance.
A4: Thank you to the reviewers for their suggestions. In 4.1 Analysis of knowledge distillation results, the epoch graph of loss and accuracy in the model training process is added, which further proves the effectiveness of knowledge distillation in improving the accuracy of lightweight networks, as follows:
During the training process Shuffv2_x0_5 loss and accuracy after individual training and ResNet18 knowledge distillation are shown in Figure 5:
|
(a) |
(b) |
Figure 5. Knowledge distillation contrasts loss and accuracy in the process of model training: (a) Loss comparison chart, (b) Accuracy comparison chart.
Q5. Provide more explanation for this sentence: Page 17, Line 211: “However, different streams need to insert CUDA events to control the execution 211 order between the Multi-Stream”.
A5: Thank you to the reviewers for their questions. In the process of programming CUDA, CUDA events are mainly responsible for synchronous Stream execution. Therefore, CUDA events must be inserted into Multi-Stream to keep different Streams synchronized, otherwise it will cause timing confusion. For easier reader understanding, it has been changed to:
When performing tasks of different streams, we must insert CUDA Events to control stream timing synchronization.
Q6. Similarly, this sentence needs to be better explained: Page 11, Line 315: “However, there will be multiple regions entering at the same time, and the time difference between the previous inference and the next inference is large”.
A6: Thank you to the reviewers for their questions. We apologize for the misunderstanding caused by my poor text description, but I have changed the content to make it easier for readers to understand:
However, in the real-time detection of photoelectric targets, the number of detection areas of different frame images differs due to the possibility of multiple targets in the same image. Multiple detection regions in the image are input into the network model simultaneously. The difference in the number of detection regions will lead to a significant difference in the inference time between the nth and the n-1 frame images. It can occur because of excessive waiting time caused by the data copy.
Q7. Determine how the hyperparameters of your deep learning model were optimized.
A7: Thank you to the reviewers for their questions. Added model parameter optimization settings in 2.2 Experimental environment, as follows:
In the network training process, the optimizer is SGD, the cross-entropy loss function is BCELoss, the momentum is set to 0.9, the initial learning rate is set to 0.001 which drops to the original 0.92 every ten generations, and the training samples for each learning are 30 for a total of 100 iterations. After completing the training, the best evaluation result is saved as the final model.
Q8. The authors mentioned that “most of the lightweight networks by knowledge distillation have higher detection accuracy than the lightweight networks trained alone.”. However, this statement is partially true because the dataset might be highly complex such that even a better network may not necessarily improve the model accuracy. In machine learning, this can refer to “Kolmogorov complexity” denoting the length of the shortest computer program that produces the output. Therefore, write a paragraph in your paper arguing that reducing the complexity of your dataset can potentially improve the accuracy of the deep learning model and reference the 2 papers listed below (that reduce the complexity of their dataset to improve the accuracy of their machine learning models)
- Bolon-Canedo, V., & Remeseiro, B. (2020). Feature selection in image analysis: a survey. Artificial Intelligence Review, 53(4), 2905-2931.
- Kabir, H., & Garg, N. (2023). Machine learning enabled orthogonal camera goniometry for accurate and robust contact angle measurements. Scientific Reports, 13(1), 1497.
A8: Thanks to the two documents provided by the reviewers, after careful reading, part of the content of the article has been modified, and these two documents have been cited and added to the references, adding the following content:
However, this is not absolute. According to "Kolmogorov complexity" reducing the complexity of the dataset can improve the accuracy of machine learning models [33, 34]. So model accuracy is determined by the complexity of the dataset and the training process.
Q10: Conclusion: Can authors highlight future research directions and recommendations? Also, highlight the assumptions and limitations (e.g., shortcomings of the present study). Besides, recheck your manuscript and polish it for grammatical mistakes (you can use “Grammarly” or similar software to quickly edit your document).
A10: Thank you to the reviewers for their suggestions. In the conclusion section, limitations on future research directions and methods are added. And use the "grammar" software to polish the manuscript and correct grammatical errors. The additions are as follows:
The entire process effectively reduces the computing power requirements of neural networks deployed in GPU computing frameworks, which is significant for designing a real-time photoelectric target detection system. The disadvantage of the method is that the hardware part is not enough accelerated and can be further improved according to the hardware characteristics in the future.
Q11. How can the results of this study be applied to other fields beyond photoelectric target detection, and what are some of the potential implications for future research in this area?
A11:In this article, the model compression and acceleration method is studied according to the requirements of photoelectric target detection. We have also tried this method in the fields of UAV detection and small target detection. At present, our research has made some progress. The methodology applies to a wide range of areas but still needs to be analyzed on a mission-by-task basis. In the future, deep learning for industrial production will face the problem of landing deployment. Actual deployment is also an important factor limiting the development of deep learning. The method pays attention to the trade-off between accuracy and real-time, has certain industrial application value, and provides a foundation for subsequent development.
Q12. What are some of the challenges faced when deploying deep learning models in GPU computing frameworks with limited computing power?
A12: I think there are two main challenges to deploying deep learning models in GPU computing frameworks with limited computing power. 1). Deep learning models rely too much on the processing performance of computing units (usually referring to graphics cards). The cost and volume of graphics cards have certain limitations on the actual deployment of the project. Therefore, it is necessary to compress and accelerate the model to achieve hardware miniaturization and cost reduction. However, the model accuracy will be lost in the compression process, and the trade-off between model accuracy and processing speed is weighed according to the actual application task. 2). The inference optimization method is not static. The inference mode of the model needs to be optimized according to the demand indicators of the deployment task. The different inference optimization methods are tried for different use scenarios, which may greatly increase labor costs.

Reviewer 2 Report
1) The need for deep learning must be clearly mentioned in the abstract.
2) The literature review must be more "critical" than just summarizing the contents.
3) Is the dataset used in this work publicly available? If yes, give the link of it.
4) The confusion matrix for each method must be given before giving the overall accuracy.
5) How many images (training and testing) are used in this work?
6) How do you validate your experimental results?
7) More inferences on the graphs shown must be given.
8) how did you choose the parameters used in the implementation process? Was it selected randomly?
9) A comparative analysis with other methods must be given.
Author Response
The reviewers are thanked for their suggestions, and the professionalism of the reviewers is seen through the suggestions. The following answers are given to the questions raised by the reviewers, and mark the article in yellow.
Q1: The need for deep learning must be clearly mentioned in the abstract.
A1: Thank you to the reviewers for their suggestions. I rewrote the abstract to add sections, as follows:
Deep learning has better detection efficiency than typical methods in photoelectric target detection. However, classical CNNs on GPU frameworks consume too much computing power and memory resources. We propose a Multi-Stream inference-optimized TensorRT (MSIOT) method to solve this problem effectively. MSIOT uses knowledge distillation to effectively reduce the number of model parameters by layer guidance between CNNs and lightweight networks. Moreover, we cooperate with TensorRT and Multi-Stream mode to reduce the number of model computation. MSIOT again increases inference speed by 9.3% based on the 4.3-7.2x acceleration of TensorRT. The experimental results show that the model's mean average accuracy, precision, recall, and F1 score after distillation can reach up to 94.20%, 93.16%, 95.4%, and 94.27%. It is of great significance for designing a real-time photoelectric target detection system.
Q2: The literature review must be more "critical" than just summarizing the contents.
A2: Thank you to the reviewers for their suggestions. In the literature review section, we have added criticism, pointed out existing problems and proposed corresponding innovations, as follows:
Ke [15] designed a fully automatic camera detection and recognition system based on the PC, which combines machine learning and neural network methods to identify surveillance camera equipment effectively. This method improved VGGNet-16, and the single forward inference time reached 5.36s, which could not meet the real-time detection requirements of photoelectric targets and was unsuitable for engineering applications. Liu et al. [16] developed a convolutional neural network photoelectric target detection and recognition system based on NVIDIA Jeston TX2, which uses a lightweight network to detect miniature indoor cameras. This method needs to improve the accuracy and inference acceleration of lightweight networks and relies too much on the computing performance of NVIDIA Jeston TX2. Moreover, Huang et al. [17] designed an improved YOLOv3 model based on the PC, which recognizes miniature cameras in a single frame. This method requires large-volume hardware support, and eliminating false targets in complex background environments needs to be more thorough, resulting in unstable detection accuracy.
Classical CNNs running on GPU compute frameworks [18-20] create problems with poor real-time performance. Therefore, this paper proposes a Multi-Stream inference-optimized TensorRT (MSIOT) inference acceleration method. This method verifies the real-time detection of photoelectric targets on the GPU computing frame with limited computing power. This paper’s significant contributions are as follows:
- In order to effectively reduce the number of model parameters and ensure the high accuracy of the neural network. We train a high-precision CNN model through knowledge distillation [24]; guided learning is performed on lightweight networks. Finally, a high-precision lightweight network model is obtained.
- In order to reduce the number of computation in the process of model inference, we deeply explore the inference acceleration principle of the TensorRT [21-23] engine based on the characteristics of the GPU computing framework and build a computational graph based on the existing network. Experiments verify the effectiveness and practicality of TensorRT inference acceleration.
- In order to solve the excessive waste of time for data replication and overlapping calculations during model inference. We optimize TensorRT to exploit CUDA (Computer Unified Device Architecture) control based on the kernel execution principle. The utilization of the GPU is fully invoked through the Multi-Stream [25] mode, further shorting the inference time of deep learning models.
Q3: Is the dataset used in this work publicly available? If yes, give the link of it.
A3: Thank you to the reviewers for their questions. The dataset used in the collaborative research decision work is temporarily not publicly available for some reason. If you need it to verify the accuracy of the experimental data of our paper. I can provide it to you.
Q4: The confusion matrix for each method must be given before giving the overall accuracy.
A4: Thank you to the reviewers for their suggestions. In section 4.2 Analysis of MNIST result add confusion matrix. The content is as follows:
We input 500 test sets of experiment 2.2 into ShuNet and ShuEng respectively for photoelectric target binary classification tasks, and the confusion matrix is identical, as shown in Figure 7:
Figure 7. Part of the test dataset photoelectric target binary classification confusion matrix.
Q5: How many images (training and testing) are used in this work?
A5: Thank you to the reviewers for their questions. This work uses a total of 13500 images, of which 6750 are true target images and 6750 are false target images, and the training set is 10800 images and the test set is 2700 images during model training.
Q6: How do you validate your experimental results?
A6: Thank you to the reviewers for their suggestions. Added 4.3 System experimental verification to verify experimental results, as follows:
4.3 System experimental verification
In order to verify the effectiveness and robustness of the above method in actual indoor scenarios, this chapter integrates a photoelectric target rapid detection system and carries out experimental system verification. Pre-place 1-2 pinhole cameras in different indoor scenarios. Hiding a pinhole camera in the corner of a complex interior or in an object that is not easy to find it difficult to identify by the human eye alone. The following experiments are carried out for the following four scenarios, with distances ranging from 3-10m, as shown in Figure 10:
Figure 10. Four different indoor experimental scenarios.
Scenario 1 is a laboratory scenario where two pinhole cameras are hidden; Scenario 2 is a conference room scenario hiding a 1.0mm diameter pinhole camera on a book; Scenario 3 simulates a 1.5mm diameter pinhole camera installed in a handbag to shoot the field of view ahead; Scenario 4 is a scenario where a 1.0mm diameter pinhole camera is hidden in a bookcase with human interference factors. The above four scenarios are challenging to find by naked eye inspection alone, and the detection results of the photoelectric target rapid detection system are shown in Figure 11:
Figure 11. The system detects the image.
The real-time detection results of the equipment are shown in Figure 12, and they are all real-time detection.
Figure 12. The system detects the results in real time.
The equipment can accurately detect pinhole cameras in different indoor environments through the above real-time detection results. It can still detect in real time with reasonable accuracy under human interference factors, proving the equipment's effectiveness and the superiority of surpassing manual troubleshooting.
While verifying the system effectiveness of the above experiments, the detection time of each scenario was counted, and the average detection time of the four scenarios was 30ms, 24ms, 33ms, and 22ms.
Scenario 3 is more complex than Scenario 2 background. The background in scenario 2 is only miscellaneous objects and some books. The background of scenario 3 contains many high-brightness metal objects, such as instruments, stools, and connecting wires. These items have specific pseudo-target characteristics. After analysis, the length of detection time depends on the number of candidate regions input. The more bright radiation in the background, the more complex the background, resulting in a sharp increase in candidate regions so that the discrimination time becomes longer.
We count the detection time in 10 different indoor scenarios, and according to the 1-second video composed of 24 frames, each detection time's average detection time is less than 42ms during the actual detection process to ensure the real-time detection video. After statistics, the average detection time of the system for ten different indoor scenarios is 32ms.
Q7: More inferences on the graphs shown must be given.
A7:Thank you to the reviewers for their suggestions. More analysis has been done on the chart content, and some of the additions are as follows:
Figure 8 and Table 4 demonstrate a significant increase in ShuffEng inference time compared to ShuffNet. The increase in the input size increases the inference time of the model. According to different input sizes, the inference time acceleration ratio can reach 4.3-7.2 ×, significantly improving the inference performance of deep learning models deployed in low-computing GPU computing frameworks. However, in the real-time detection of photoelectric targets, the number of detection areas of different frame images differs due to the possibility of multiple targets in the same image.
Multiple detection regions in the image are input into the network model simultaneously. The difference in the number of detection regions will lead to a significant difference in the inference time between the nth frame image and the n-1 frame image. It can occur because of excessive time waiting caused by the copy of the data. To solve this problem, we use Multi-Stream to tuning TensorRT and conduct comparative experiments.
Q8: how did you choose the parameters used in the implementation process? Was it selected randomly?
A8:Thank you to the reviewers for their suggestions. During implementation, parameters are not selected randomly. Add parameter configuration in 2.2 Experimental environment, as follows:
In the network training process, the optimizer is SGD, the cross-entropy loss function is BCELoss, the momentum is set to 0.9, the initial learning rate is set to 0.001 which drops to the original 0.92 every ten generations, and the training samples for each learning are 30 for a total of 100 iterations. After completing the training, the best evaluation result is saved as the final model.
Q9: A comparative analysis with other methods must be given.
A9:Thank you to the reviewers for their suggestions. In the 4.3 System experimental verification section, a comparative analysis with other methods has been added, as follows:
We count the detection time in 10 different indoor scenarios, and according to the 1-second video composed of 24 frames, each detection time's average detection time is less than 42ms during the actual detection process to ensure the real-time detection video. After statistics, the average detection time of the system for ten different indoor scenarios is 32ms, and the performance of similar methods is compared, as shown in Table 5.
Table 5. Comparison of TP, recall, and average Time with different methods
|
Method |
TP |
Recall |
Average Time |
|
Ke[15] |
- |
75.5% |
5.36s |
|
Liu[16] |
466(500) |
93.2% |
166ms |
|
Huang[17] |
189(200) |
94.5% |
104ms |
|
MSIOT(Ours) |
477(500) |
95.4% |
32ms |
From Table 5, it can be concluded that MSIOT is better than the method proposed by Ke [15] and Liu [16] in terms of recall, which is the same as the YOLOv3-4L method proposed by Huang [17]. MSIOT model compression and acceleration are much shorter in detection time than the other three methods. MSIOT can prove the system detection function's superior processing speed and real-time performance.

Reviewer 3 Report
The paper presents an inference-optimized high-performance photoelectric target detection method to improve real-time performance. Mutli-Stream pattern is implemented by combining knowledge distillation and TensorRT. The experimental results demonstrate that the prediction accuracy of the lightweight network can be improved using knowledge distillation. The accelerated inference of the lightweight network Shuffv2_x0_5 on the NVIDIA Jeston Nano is also validated using TensorRT and MSIOT. Overall, the idea is interesting and has practical application value. I have other comments as follows.
1. In this paper, knowledge distillation is adopted to improve the performance of the model. However, the rationale for knowledge distillation being used to improve model performance is not explored in the introduction.
2. In Fig. 5, photoelectric target detection results are too small to see clearly. The authors can add the close-ups for each result.
3. In Table 3, the results of ShuffNet and ShuffEng are the same, please explain the results.
In a word, the paper can be considered for publication after a minor revision.
Author Response
The reviewers are thanked for their suggestions, and the professionalism of the reviewers is seen through the suggestions. The following answers are given to the questions raised by the reviewers, and mark the article in yellow.
Q1:In this paper, knowledge distillation is adopted to improve the performance of the model. However, the rationale for knowledge distillation being used to improve model performance is not explored in the introduction.
A1: Thank you to the reviewers for their suggestions. We have added the basic principles of knowledge distillation to the introduction, as follows:
In order to effectively reduce the amount of model parameters and ensure the high accuracy of the neural network. We train a high-precision CNN model through which knowledge distillation [24] guided learning is performed on lightweight networks. Finally, a high-precision lightweight network model is obtained.
And Detailed in section 3.1 Knoledge distillation, as follows:
The complete knowledge distillation concept was first proposed in 2014 by Google Labs Hinton [24], which experimentally verified its feasibility and the effectiveness of CNN compression on MNIST data sets. The probability of an error class is relatively small in the probability distribution output of a well-trained photoelectric target detection model. Since its relative probability distribution hides the feature information that the real label does not have, knowledge distillation is introduced to improve the discriminant accuracy of lightweight networks. As shown in Figure 2:
Figure 2. Knowledge distillation schematic diagram.
The temperature coefficient is added to the output layer of Softmax to smooth the probability distribution of the network’s output. The output obtained is called a soft target, with soft targets and real tags work together to guide student network training. The loss function can be expressed as:
Neural networks typically generate class probabilities using a Softmax output layer, which normalizes to probability . Besides, represents the cross-entropy between the predicted output of the student network and the true label, and is a hyperparameter that adjusts the proportion between the predicted output after smoothing by the student network and the teacher network. When cross-entropy is backpropagated, the gradient changes to the original , which is smoothed by a hyperparameter . Therefore, to preserve the scale of its gradient consistent with the scale of the cross-entropy corresponding to the true label. It is necessary to multiply the smoothed cross-entropy by .
Q2:In Fig. 5, photoelectric target detection results are too small to see clearly. The authors can add the close-ups for each result.
A2:Thank you to the reviewers for their suggestions. Sorry for the bad visual experience. The results of Figure 6 (original Figure 5) have now been magnified,as follows:
Figure 6. Photoelectric target detection forward inference comparison test: (a) active image, (b) inference result of ShuffNet, and (c) inference result of ShuffEng.
Q3:In Table 3, the results of ShuffNet and ShuffEng are the same, please explain the results.
A3: Thank you to the reviewers for their questions. We added clarification to the 4.2 MSIOT results analysis, as follows:
ShuffNet inference weight precision is 32 bits. ShuffEng inference full precision is also 32 bits. It is fully verified through Table 3 that ShuffNet and ShuffEng have no impact on the Accuracy, Precision, Recall, and F1 score indicators of photoelectric target detection when the inference weight type is the same and only affect the inference speed of the model.

Round 2
Reviewer 1 Report
Comments are addressed.
Reviewer 2 Report
The article is improved. It can be accepted now.